# Navigation Guidance for Percutaneous Splanchnic Nerve Radiofrequency Neurolysis: Preliminary Results

**DOI:** 10.3390/medicina58101359

**Published:** 2022-09-28

**Authors:** Stavros Grigoriadis, Dimitrios Filippiadis, Vasiliki Stamatopoulou, Efthimia Alexopoulou, Nikolaos Kelekis, Alexis Kelekis

**Affiliations:** 2nd Department of Radiology, National and Kapodistrian University of Athens, “Attikon” University Hospital, 12462 Athens, Greece

**Keywords:** navigation, computed tomography, pain, neurolysis

## Abstract

*Background and Objectives*: To describe preliminary results upon the application of the “Cube Navigation System” (CNS) for computed tomography (CT)-guided splanchnic nerve radiofrequency neurolysis. *Materials and Methods*: CT-guided splanchnic nerve neurolysis was performed in five patients; in all cases, neurolysis was performed under CT guidance using the CNS. The mean patient age was 71.6 years (range 54–81 years; male/female: 5/0). Technical success, parameters of the neurolysis session and complications were evaluated. Technical success was defined as a needle position on the defined target. Session parameters included procedure time and number of scans. The CIRSE reporting system was used for complications’ classification and grading. *Results*: Technical success was obtained in all cases; in 1/5 patients, a slight correction in needle orientation was necessary. Mean procedure time was 12.4 min (range 8–19 min); an average of four CT scans was recorded in the five neurolysis sessions. There were no complications or material failures reported in the present study. *Conclusions*: Preliminary results of the present study show that computed tomography (CT)-guided splanchnic nerve radiofrequency neurolysis using the CNS is an accurate and time-efficient percutaneous procedure. More prospective and comparative studies with larger patient samples are necessary for verification of this system as well as for drawing broader conclusions.

## 1. Introduction

Percutaneous imaging-guided interventions, e.g., radiofrequency ablation, are established in clinical routine for numerous diagnostic and therapeutic indications. Due to its high spatial resolution, fast volume acquisition, ability to capture detailed images of varying types of body tissue, broad availability and relative low cost, computed tomography (CT) is frequently used as for imaging-guidance. Accurate needle positioning is essential for the safety and efficacy of CT image-guided percutaneous interventions, as deviations from the pre-defined target of the needle can lead to lengthy procedures, unnecessary radiation exposure, adverse events, or even treatment failures [1]. In the vast majority of centers, needle positioning during CT-guided percutaneous techniques is performed manually, i.e., free-hand. With this method, deviation from planned to actual needle insertion angle is approximated by the physician, with the result that accuracy is highly dependent on the experience as well as visual and spatial abilities of the physician. The free-hand technique may be sufficient when performing basic in-plane interventions, but in case of out-of-plane trajectories, some access routes prove to be difficult to estimate. To address this problem, various CT navigation guidance systems, including systems based on optical and electromagnetic inputs, have been developed [2]. However, these systems have been limited in their adoption due to different reasons including cost, longer duration of the procedure and increased workflow complexity.

Recently, a new needle guidance system, the “Cube Navigation System” (CNS), was introduced (Figure 1). The CNS is a small, cube-shaped, patient-mounted, self-adhesive apparatus with multiple through holes in the upper and lower surface. The accompanying software recognizes the cube in the planning scan and, with use of a virtual needle, allows the physician to plan an optimal trajectory to the target. The route is displayed as coordinates on both the top and bottom grid of the cube. The physical puncture needle is then inserted into the respective holes on the PC. In an in vitro phantom study, the CNS showed significantly improved accuracy, lower procedure time and an overall lower intervention time compared to punctures performed with the FHM [3].

In this technical report, we describe our initial clinical experiences with the CNS. For this study, the puncture cube (PC) containing openings for needles 18–22 G was used. The system was applied in a small series of five patients referred for radio-frequency ablation of the splanchnic nerves.

## 2. Patients and Methods

### 2.1. Patients

All patients were informed about the technique and potential complications of the splanchnic nerve RF ablation and written informed consent was obtained. All patients had undergone a test block of their splanchnic nerves to ensure improvement in their pain levels prior to their procedure. The review board of our institution approved the study and the principles of national legislation and the Declaration of Helsinki were followed. Inclusion criteria included adult pancreatic cancer patients suffering from intractable pain refractory to conservative analgesic medication; in all patients, coagulation parameters should have been within normal limits whilst life expectancy should be >3 months. Exclusion criteria included patients not consenting to the study and/or the technique, uncorrectable coagulopathy, infection, and life expectancy <3 months. Laboratory work-up, including blood count and coagulation testing, was performed in all patients the morning of the intervention. Pain scores using the visual analog scale were recorded prior to the intervention and on the morning following the intervention.

### 2.2. Cube Navigation System

The “CNS” (Medical Templates AG, Egg, Zurich, Switzerland) was described recently in detail in an ex vivo study [3]. In short, the PC is a disposable cube-shaped apparatus consisting of an upper- and a lower-grid template, with fields marked both with letters and numbers (e.g., field B2). This cubic-shape apparatus will collapse during a puncture if the needle is too short, or if the needle path needs to be adjusted. The pre-mounted adhesive on every foot of the cube allows for fixation on the skin of the patient. The PC consists of polypropylene, which is hyperdense on CT planning scans (approx. −200 Hounsfield units) (Figure 2).

The CT images are sent to the dedicated software “Synedra View Professional” (Synedra information technologies GmbH, Innsbruck, Austria) via PACS. The software recognizes the location of the PC on the patient’s skin, and a virtual model of the cube is created. Then, 3D MPR reconstructions and a virtual needle path are used by the operator for selection of target approach through the PC. Once a path is chosen, the corresponding holes on both the upper and lower surfaces are indicated by green dots on the virtual cube, as well as in the 3D MPR view. As the virtual needle has distance markers, using puncture needles with distance indicators allow for control of the puncture depth (Figure 3A). To achieve accuracy in the insertion of the puncture needle, the software automatically places the path of the virtual needle in one corner of the indicated hole. Next, the operator introduces the puncture needle into the determined coordinates on both layers of the PC. Sequential control scans are performed until the needle reaches the target.

### 2.3. Puncture Description

Patients were placed in prone position on the CT table. Prior to radio frequency ablation all patients were sedated with incrementally increasing i.v doses of fentanyl and propofol. Vital signs were monitored throughout the procedure, including oxygen saturations and ECG. Computed tomography sequential scans (120 Kv, 240 mAs wavelength and 2 mm slice thickness) were performed for planning, targeting and intra-procedural modification. Based on a lateral and an antero-posterior topogram at the 12th thoracic vertebral body (TH12), the approximate puncture site was marked on the skin. After skin disinfection, the PC was placed over the approximate puncture site and a CT planning scan was obtained during breath hold of the patient. The images were transferred to the software as described above and the access route to the target, antero-lateral on the TH12 vertebral body, was determined. The coordinates to introduce the needle were recorded for the upper and lower plate (Figure 3B). A 15 cm long, 20–gauge radiofrequency needle with a 10 mm active tip (Diros OWL^®^ RF Probe, Diros Technology Inc., Ontario, ON, Canada) was inserted and its approach until the level of splanchnic nerve plexus antero-lateral to the vertebral body was evaluated with sequential CT scans until the designated target was reached. The cube was collapsed, as designed, after the needle was firmly introduced a few centimeters into the patient. The neurolysis session was performed with an ablation protocol according to the manufacturer’s guidelines (two circles of 90 s at a temperature between 80–90 °C) (Figure 4A). Patients remained in the hospital overnight before discharge. All five procedures were performed by two highly experienced interventional radiologists who had no prior experience with the CNS.

### 2.4. Analyzed Items

Recorded were: procedure time (defined as time from the first CT topogram to the last CT control); number of scans (first control scan after needle introduction to scan when final position of the needle was reached); correction of needle path through the same cube hole after first necessary introduction of needle; in-plane or out-of-plane access (along *z*-axis of the scanner); complications (classified and graded according the CIRSE reporting system [4]); change in pain score (VAS).

## 3. Results

Technical success (i.e., positioning of the needle on the desired location) was obtained in all cases. All five patients treated received systematic therapy during the neurolysis and study period. Mean patient age was 71.8 years (range 54–81 years; all male). The mean procedure time to reach the target, as calculated from the start of the planning scan with the cube, was 12.4 min (range 8–18 min). An average of four control scans was used to reach the final target. In one patient, the first control scan required a correction of needle path after the first introduction of the needle, as the needle path was slightly misaligned to reach the target, most likely due to a breathing artifact. In all patients, a slight out-of-plane access was chosen with respect to the z–axis of the scanner. No complications occurred. The median ablation temperature was 85 °C. Mean pain score prior to neurolysis was 9.8 NVS units (range 9–10); at 3 months post-neurolysis, the mean pain score was 1.2. NVS units (range 0–3). There were no complications or material failure reported in the present study.

## 4. Discussion

In this small series of patients, the CNS was a promising guidance tool for percutaneous CT-guided interventions. The system was easy to install, simply requiring installation of software on a standard computer connected to the CT scanner and attaching the PC to the patient. Usage of the system was simple, and needle placement proved overall to be highly accurate. In only one of the five punctures was the initial needle position slightly incorrect, most likely due to some breathing artifact or to the operators being early in the learning curve.

Time requirements for correct needle placement with the free-hand technique can be quite variable in our experience. This variability seemed to be reduced considerably for use of the CNS, which might increase ease of CT-guided interventions. It can furthermore be assumed that the safety and efficacy of this technique could increase by the use of double-oblique trajectory routes without compromising accuracy. In fact, in the presented cases, the system allowed for an out-of-plane access route, thus achieving optimal needle placement. However, these assumptions need to be analyzed in a larger prospective study.

In our initial experience, four scans were used on average to control needle progression and final position. One can presume that, with increased experience and confidence in the system, the number of scans could be reduced even further.

Numerous other navigation systems are available [5,6,7,8]. In our view, the CNS has several advantages over these systems. First, the system does not need direct visibility between fiducials and the tracking camera, as is necessary for optical tracking systems. Second, no interferences which might impair the utility of magnetic tracking systems occur, and there is no need for proximity to a magnetic field generator, which can be cumbersome in a small operating space. Third, slight movements of the patient after the scan do not result in inaccuracies as with laser-guidance systems. Fourth, electromagnetic or optical tracking–based navigation systems do not allow standard needles to be used, whereas the CNS does. Finally, the simple installation of the CNS means that associated installation costs remain very low.

In contrast with other systems, the CNS offers stability of the needle during image acquisition (Figure 4B) as well as guidance. This could be of interest especially in superficial lesions as it is held in place by the cube holes. Repositioning of the needle and/or correction of the angulation can be performed when the upper surface collapses over the lower one after needle insertion. In summary, given the ease of installation and use of the CNS, as well as the high degree of accuracy attainable with the system, we consider the CNS an advantageous addition to routine clinical practice. Improvements such as larger through holes and removable plates after needle insertion provide CNS with the potential to be widely applied for a variety of clinical indications.

There are some limitations of the CNS. First, the PC has a size restriction for needle sizes of up to 18 gauge; thus, commonly performed radio-frequency procedures that require larger needles cannot be enacted. Second, although corrections of the needle path still remain possible after needle insertion, the ease of needle manipulation is impaired by the PC. Third, firmly attaching the PC to certain skin locations can be challenging. Limitations of the study itself include the small number of patients and the lack of comparison to a group of patients who underwent free-hand neurolysis under CT guidance.

## 5. Conclusions

The use of the CNS proved to be simple, and punctures with the CNS were, in this initial case series, accurate. The CNS allowed for good calculation of a double-oblique access route, allowing the user to choose the optimal pathway, and seemed to reduce variability in procedure time. Prospective and comparative studies with larger patient samples for different indications and access routes are necessary for a more thorough evaluation of this system.

## Figures and Tables

**Figure 1 medicina-58-01359-f001:**
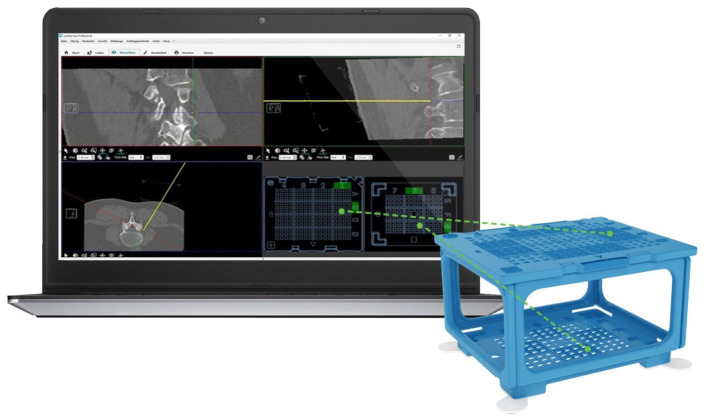
The CNS system consisting of the PC on the right, and software on the left. The software displays the standard MPR views in coronal, sagittal and axial views. Shown here is an example infiltration at the lumbar level. The chosen trajectory of the virtual needle is shown as a highlighted yellow cross-hair marker which can be freely moved. Based on the trajectory of the needle, the corresponding coordinates in the upper-left and lower-right templates are highlighted by the software. The physical puncture needle is then inserted through these holes.

**Figure 2 medicina-58-01359-f002:**
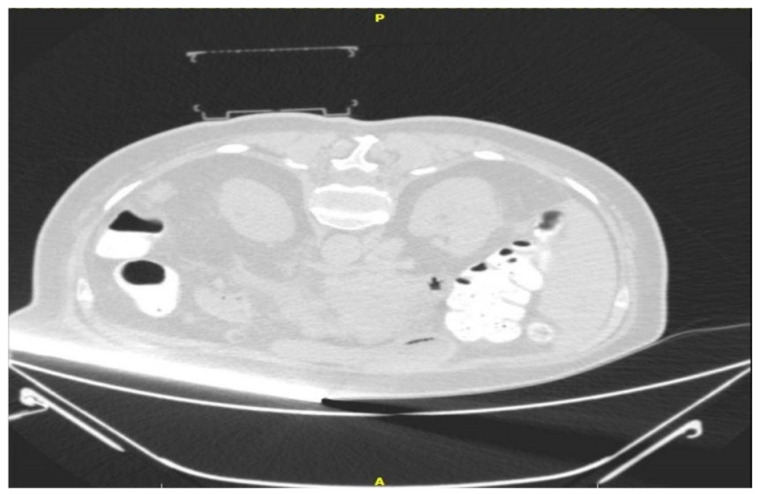
Initial CT scan with target and the cube mounted on the skin.

**Figure 3 medicina-58-01359-f003:**
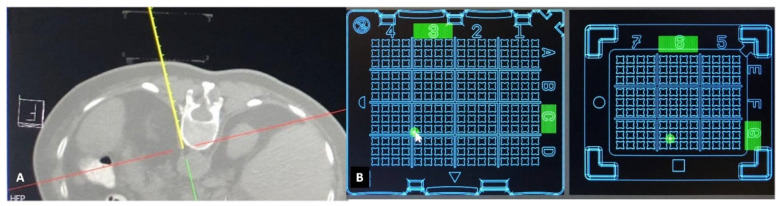
Planning of the puncture: (**A**): The yellow line represents the planned puncture path via a virtual needle (markers on the line represent cm). (**B**): The software calculates the entry (upper plate left) and exit point (lower plate right) and displays green dots at the coordinates to be used for this particular planned line.

**Figure 4 medicina-58-01359-f004:**
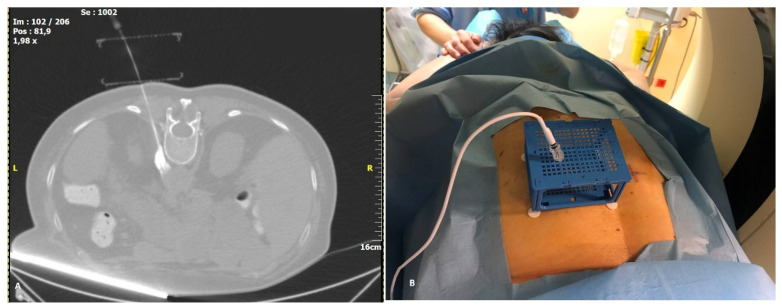
Final image prior to treatment: (**A**): Corrected (MPR) CT image of the cube with the needle in place, showing the whole needle trajectory. At the tip of the needle, there is hyperdense contrast pooling, verifying the correct needle position. (**B**): Actual image of the needle and electrode in place inside the cube.

## Data Availability

Data will be provided by the authors upon reasonable request.

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
