# Peer review of "Navigation Guidance for Percutaneous Splanchnic Nerve Radiofrequency Neurolysis: Preliminary Results"

_medicina, 2022, doi:10.3390/medicina58101359_

Round 1

Reviewer 1 Report

Dear Author;

1. In order to claim that the study reduces the variability in procedure time, it should have its own control group. Data on the time in the studies mentioned in the discussion should be shared if this is not possible.

2. A similar problem exists with complications.

Author Response

Point 1: Dear Author;

  1. In order to claim that the study reduces the variability in procedure time, it should have its own control group. Data on the time in the studies mentioned in the discussion should be shared if this is not possible.

Response 1: We thank reviewer one for his/her comments.  We have included the recorded duration of the procedure (mean +_ SD) however we feel that providing time data of mentioned studies does not add to the value of the current submission since no nurolysis procedures have been included there. We have added in the limitations paragraph the following: “Limitations of the study itself include the small number of patients  and  the lack of comparison to a group of patients who underwent free hand neurolysis under CT-guidance.”

Point 2: 2. A similar problem exists with complications.

Response 2: We thank reviewer one for his/her comments.  We have added in the limitations paragraph the following: “Limitations of the study itself include the small number of patients  and  the lack of comparison to a group of patients who underwent free hand neurolysis under CT-guidance.”

Reviewer 2 Report

Thank you for your paper. The content must be improved.

The sample size is limited and conditions the results and impact of the study.

What differences or advantages can this device really bring? Could the results be easily transferred to clinical practice?

Author Response

Point 1:  Thank you for your paper. The content must be improved.

The sample size is limited and conditions the results and impact of the study.

Response 1: We thank reviewer one for his/her comments.  We have included the recorded duration of the procedure (mean +_ SD) however we feel that providing time data of mentioned studies does not add to the value of the current submission since no nurolysis procedures have been included there. We have added in the limitations paragraph the following: “Limitations of the study itself include the small number of patients  and  the lack of comparison to a group of patients who underwent free hand neurolysis under CT-guidance.”

Point 2: What differences or advantages can this device really bring? Could the results be easily transferred to clinical practice?

Response 2: We thank reviewer one for his/her comments.  We have added in the discussion section  the following: “In summary, given the ease of installation and use of the CNS, as well as the high degree of accuracy attainable with the system, we consider the CNS an advantageous addition to routine clinical practice. Improvements such as larger through holes and removable plates after needle insertion provide CNS with the potential to be widely applied for a variety of clinical indications.”

Round 2

Reviewer 1 Report

Dear Author;

Thanks for your comments and corrections.

Reviewer 2 Report

Thank you for your comments and the corrections made.

I recommend accepting the article after minor revision.